# Study on Production Technology and Volatile Flavor Analysis of Fragrance *Zanthoxylum* Seasoning Oil

**DOI:** 10.3390/foods12112173

**Published:** 2023-05-27

**Authors:** Hang Li, Jingxuan Sun, Xinyi He, Chenyun Zhang, Zhenyu Liao, Dong Li, Hongbin Wang

**Affiliations:** 1College of Food Science and Biotechnology, Tianjin Agriculture University, Tianjin 300392, China; lihang18888@163.com (H.L.);; 2Tianjin Engineering and Technology Research Center of Agricultural Products Processing, Tianjin 300392, China; 3PONY Testing Technology (Tianjin) Co., Ltd., Tianjin 300392, China; 4Tianjin Honglu Foods Co., Ltd.,Tianjin 301713, China

**Keywords:** *Zanthoxylum* seasoning oil, cold pressing, volatile flavor compounds, Maillard reaction

## Abstract

Dried green pepper and first-grade extracted soybean oil were selected as raw materials to study the effect of the Maillard reaction and cold-pressed compound on the quality of *Zanthoxylum* seasoning oil and its aroma-enhancing effect. The results showed that the optimal technology was as follows: the ratio of material to liquid was 1:5, the heating temperature was 110 °C, the reaction time was 25 or 30 min, and the addition of reducing sugar was 2%. The optimum ratio of fragrant *Zanthoxylum* seasoning oil was 1:7 for cold pressing oil and hot dipping oil. Compared with *Zanthoxylum* seasoning oil, it is based on the Maillard reaction and had a more intense and persistent aroma. The taste of fragrant *Zanthoxylum* seasoning oil was the best of the three blended oils. The possible types of volatile flavor compounds in the three kinds of *Zanthoxylum* seasoning oils detected by Heracles II ultra-fast gas phase electronic nose were, respectively, 16, 19, and 15. Among the three kinds of *Zanthoxylum* seasoning oils, the content of limonene, linalool, Eucalyptol, n-pentane α-Pinene, myrcene, and phellandrene was more abundant, which indicated that olefins and alcohols contributed more to the overall flavor of the three kinds of *Zanthoxylum* seasoning oils.

## 1. Introduction

*Zanthoxylum bungeanum* originated in China [1], belonged to Rutaceae Zanthoxylum Linn. and was one of the main condiments in China. Zanthoxylum seasoning oil, as one of the processed products of *Zanthoxylum bungeanum*, is popular among consumers [2]. As a seasoning oil, Zanthoxylum seasoning oil should have a good flavor while meeting the standards of vegetable oil. Therefore, volatile flavor compounds were also an important criterion to evaluate the quality of Zanthoxylum seasoning oil. The Heracles II Rapid Gas Phase electronic nose is a new type of gas flavor analysis system [3]. The processing speed is fast and simple and the cost is optimal [4]. It is widely used to identify volatile flavor substances in fruits, grains, milk and vegetables [5,6,7].

Most of the Zanthoxylum seasoning oil sold in the market today was made by directly soaking peppers in vegetable oil. However, in the previous research on the processing of Zanthoxylum seasoning oil, it was found that the numb taste and aroma components would be partially lost during its processing and storage [8], both of which were important indicators for the quality evaluation of Zanthoxylum seasoning oil. Therefore, how to increase the aroma of Zanthoxylum seasoning oil and its persistence of numb taste was particularly important. At present, apart from the oil immersion method, the main methods for extracting Zanthoxylum oil from Zanthoxylum include microwave-assisted extraction [9], ultrasonic assisted extraction [10], and supercritical CO_2_ extraction. However, the operation of microwave-assisted extraction is relatively complex, and the ultrasonic-assisted extraction method requires the addition of organic solvents, which can easily cause solvent residues. The supercritical CO_2_ extraction method has high safety, no solvent residues, and less loss of flavor substances in Zanthoxylum, but the cost is high. In order to increase the aroma of Zanthoxylum seasoning oil, the experiment mainly adopts two ways, one is to add a Maillard reaction, and the other is to use cold pressing technology.

The essence of the Maillard reaction is the reaction between a carbonyl group and the amino group [11]. It was a kind of complex reaction that can produce a variety of different flavor compounds in various ways [12,13]. The Maillard reaction can be divided into three stages. The first stage was the condensation reaction between reducing sugars and amino acids. The second stage was an Amadori or Heyns rearrangement. The final stage mainly involved flavor formation, in which amino compounds undergo a series of reactions (e.g., decomposition, cyclization, polymerization, etc.) to form a series of aromatic compounds [14,15]. These aromatic compounds constitute the characteristic flavors produced by the Maillard reaction.

In the process of the Maillard reaction, different substrates and catalytic conditions will produce different flavors, which are widely used in food processing and industrial production, mainly involving flavor, vegetable meat, seasoning bases, etc. In addition, the Maillard reaction is also widely used in oil processing, such as fragrant rapeseed oil [16,17], fragrant sunflower oil [18], fragrant sesame oil [19], etc. However, there was no report on the preparation of fragrant Zanthoxylum seasoning oil based on the Maillard reaction.

In addition to the Maillard reaction, cold pressing was also a new method for preparing Zanthoxylum seasoning oil, which could retain the volatile flavor compounds in *Zanthoxylum bungeanum* very well. In the cold pressing method, Zanthoxylum seasoning oil was usually made from fresh pepper by pressing at low temperatures (≤65 °C) [20]. However, there is no report on the cold pressing technology of Zanthoxylum seasoning oil with dry pepper as the raw material.

In this study, the enzymatic hydrolyzate and reducing sugar were used as raw materials, using sensory evaluation, peroxide value, acid value and fagaramide as indicators to explore the effects of heating temperature, reaction time, material-to-oil ratio and reducing sugar addition on Zanthoxylum seasoning oil based on Maillard reaction. Moreover, single-factor experiments and orthogonal tests were used to optimize the technology of Zanthoxylum seasoning oil based on the Maillard reaction. Cold pressing oil and hot dipping oil were mixed in different proportions to determine the optimal ratio of fragrant Zanthoxylum seasoning oil. The Heracles II ultra-fast gas phase electronic nose was applied to test the volatile flavor compounds of Zanthoxylum seasoning oil based on Maillard reaction, fragrant Zanthoxylum seasoning oil and Zanthoxylum seasoning oil, to determine the flavor enhancement effect of Maillard reaction and cold pressed compound on Zanthoxylum seasoning oil, and to provide some theoretical basis for the product development of Zanthoxylum seasoning oil manufacturers.

## 2. Materials and Methods

### 2.1. Materials

Dried green pepper: from Yunnan (moisture content 9.6%, protein content 6.81 g/100 g); First grade extracted soybean oil: Shandong Yuwang Ecological Food Co., Ltd. (Yucheng, China) (acid value 0.16 mg/g, peroxide value 0.08 g/100 g).

Glacial acetic acid, isopropanol, ether: analytical pure; n-Hexane: chromatographic pure, Tianjin Tianli Chemical Reagent Co., Ltd.; Phenolphthalein indicator, methanol, potassium iodide: analytical pure, Tianjin Damao Chemical Reagent Factory; Trichloromethane: analytical pure, Tianjin Chemical Reagent Supply and Marketing Company; Soluble starch, 0.1 mol/L *NaOH standard titrant*, 0.01009 mol/L Na_2_S_2_O_3_ standard titrant: Tianjin No. 3 Chemical Reagent Factory; Hydroxy-β-sanshool: Shanghai Yuanye Biotechnology Co., Ltd. (Shanghai, China).

### 2.2. Instruments and Equipment

HZ-K500 C electronic balance: Shanghai Youke Instrument Co., Ltd. (Shanghai, China); Heracles II ultra-fast gas phase electronic nose: Alpha M.O.S, France; UV-800 Vwas spectrophotometer: Japan Hmadzu Company; CM-5 colorimeter: Shenzhen Sanenzhi Technology Co., Ltd.; Agilent 7890 A gas chromatograph: American Agilent Company; E2695-UV2475 high-performance liquid chromatograph: American Waters Company.

### 2.3. Methods

#### 2.3.1. The Preparation of Zanthoxylum Seasoning Oil Based on the Maillard Reaction

A certain amount of reducing sugar was weighed and added to the enzymatic hydrolyzate of degreasing *Zanthoxylum bungeanum*. After shaking well, it was mixed with *Zanthoxylum* seasoning oil in proportion and co-heated for a period of time and then cooled naturally to room temperature. The mixed solution was centrifuged and the upper layer of the oil was taken, which was *Zanthoxylum* seasoning oil based on the Maillard reaction and marked as MH.

#### 2.3.2. The Preparation of Fragrant *Zanthoxylum* Seasoning Oil

The processing technology of fragrant *Zanthoxylum* seasoning oil was as follows. Weigh the same amount of oil and ground prickly ash, mix and soak for 4 h, and then obtain the cold pressed oil (LZ) and prickly ash cake by hydraulic press. *Zanthoxylum* prickly ash cake as raw material, hot vegetable oil was added, and hot immersed oil (RJ) was obtained through plate and frame filtration. Moreover, cold-pressed oil (LZ) and hot immersed oil (RJ) were mixed in proportion to obtain rich flavored *Zanthoxylum* prickly ash seasoning oil. The process flow chart of fragrant Zanthoxylum seasoning oil was shown in Figure 1.

The preparation of cold pressing oil: 4500 g dried green pepper was weighed and crushed through a 10 mesh sieve and poured into an extraction tank containing 4500 g soybean oil. Then the mixture was stirred every 5 min to make them fully mixed. After mixing three times of mixing, it was covered and soaked for 4 h before being pressed in a hydraulic oil press to obtain cold-pressed oil, marked as LZ.

Preparation of hot dipping oil: The pressed *Zanthoxylum bungeanum* cake was put into the extraction tank. The soybean oil was heated to 180 °C and poured into the extraction tank (*Zanthoxylum bungeanum* cake: soybean oil 1:9). The two were fully reacted by constant stirring, and then naturally cooled to room temperature to obtain hot dipping oil, which was marked as RJ.

Preparation of different ratios of Zanthoxylum seasoning oil: The cold pressing oil and hot dipping oil were mixed in different ratios to obtain *Zanthoxylum* seasoning oil with ratios of 1:1, 1:2, 1:3, 1:4, 1:5, 1:6, 1:7, 1:8, 1:9 and 1:10, respectively, (Total volume of oil per serving was 100 mL). After the Quality Indexes, sensory evaluation and flavor analysis, the optimum ratio of *Zanthoxylum* seasoning oil was defined as fragrant *Zanthoxylum* seasoning oil, which was marked as LH.

#### 2.3.3. Single-Factor Experiment of *Zanthoxylum* Seasoning oil Based on Maillard Reaction

The effect of material-to-oil ratio on sensory evaluation; 1.2 g glucose was weighed and dissolved in the mixture with material-to-oil ratios of 1:1, 1:3, 1:5, 1:7 and 1:9 (the total weight of enzymatic hydrolysate and *Zanthoxylum* seasoning oil was 48 g), which then were reacted at 120 °C for 30 min to investigate the effect of the ratio on the sensory evaluation of *Zanthoxylum* seasoning oil.

The effect of heating temperature on sensory evaluation and quality;1.2 g glucose was weighed and dissolved in 8 g enzymatic hydrolysate. Then 40 g *Zanthoxylum* seasoning oil was added and reacted at 100, 110, 120, 130 and 140 °C for 30 min to investigate the effects of different heating temperatures on the sensory and quality indexes of *Zanthoxylum* seasoning oil.

The effect of reaction time on sensory evaluation and quality; 1.2 g glucose was weighed and dissolved in 8 g enzymatic hydrolysate. Then 40 g *Zanthoxylum* seasoning oil was added and reacted at 120 °C for 15, 20, 25, 30, 35 min to investigate the effects of different reaction temperatures on the sensory and quality indexes of *Zanthoxylum* seasoning oil.

The effect of reducing sugar addition on sensory evaluation; 0.48, 0.72, 0.96, 1.2, and 1.44 g reducing sugar were weighed and dissolved in 8 g enzymatic hydrolysate. Then 40 g *Zanthoxylum* seasoning oil was added and reacted at 120 °C for 30 min to investigate the effect of the addition on the sensory evaluation of *Zanthoxylum* seasoning oil.

#### 2.3.4. Sensory Evaluation

(1)Selection of evaluators: The study was reviewed and approved by the Tianjin Agricultural University IRB and informed consent was obtained from each subject prior to their participation in the study. The research group conducted strict screening of recruited evaluators including bitter sensitivity and signed informed consent before the experiment. The inclusion and exclusion criteria of evaluators refer to the literature method. The inclusion criteria are ① Age of 18~40 years old and healthy; ② Ensure the participation of oral evaluation time; ③ Have subjective initiative (willingness and interest); ④ Read and fully understand the subject’s instructions and sign the informed consent. Exclusion criteria: ① Patients with severe oral diseases; ② The mentality is too nervous; ③ Those who ate (especially stimulant food) 3 h before tasting the solution; ④ Pregnant, lactation, genetic history, allergy to the sample to be evaluated, cholecystitis and gastrointestinal diseases; ⑤ Other adverse tastes that may affect the taste, such as smoking, drinking, etc.; ⑥ Abnormal taste function.(2)Invite professional evaluation experts and ordinary consumers according to GB/T 16291.1-2012 “Sensory Analysis Selection, training and management of evaluators general guidelines Part 1: Selection of evaluators”, GB/T 16291.2-2010 “Sensory analysis Selection, training and management of Evaluators General guidelines Part 2: Expert Evaluators”. Ten evaluators were selected as subjects to evaluate Zanthoxylum flavoring oil by aroma, taste and color. Standardized training: Referring to DBS51/008-2019, an appropriate number of samples were taken in the beaker and the color was observed under natural light, the product color was evaluated according to Table 1. Heat a beaker in a water bath to 50 °C, and stir it quickly with a glass rod, the product odor was evaluated according to Table 1. Rinse your mouth with warm water, and taste the taste of the product was evaluated according to Table 1 and the taste interval of each product was 5 min.

#### 2.3.5. Measurement of Indicators

Acid value: According to GB 5009.229-2016 *Determination of Acid Value in National Food Safety Standard*; Peroxide value: According to GB 5009.227-2016 *Determination of Peroxide Value in National Food Safety Standard*; Analysis of fatty acid compositions: According to GB 5009.168-2016 *Determination of Fatty Acids in National Food Safety Standard*, normalization method was implemented; Vitamin E: According to GB 1886.233-2016 *Food Safety National Standard—Food additive—Vitamin E*; Sterol: According to GB/T 25223-2010 *Animal and Vegetable Fats and Oils—Determination of Individual and Total Sterols Contents—Gas Chromatographic Method.*

#### 2.3.6. Determination of Amide Compounds

The fagaramide content was determined according to the method in Appendix A of DBS51/008-2019 *Local Food Safety Standard—Zanthoxylum seasoning oil*. The content of hydroxy-β-sanshool was calculated according to the standard curve, and the regression equation was y = 0.1501x − 0.1288 (*R*^2^ = 0.9983), where y was the absorbance value and x was the fagaramide content.

#### 2.3.7. Determination of Volatile Flavor Compounds

The volatile flavor compounds of oil samples from the Maillard reaction (MH), fragrant Zanthoxylum seasoning oil (LH) and *Zanthoxylum* seasoning oil (HL) of the volatile flavor compounds were taken by Heracles II ultra-fast gas phase electronic nose. Briefly, 2 g of the oil sample was placed in a 20 mL headspace bottle and water bath for 24 min (55 °C) 5000 μL was extracted with an injection needle gas, and injected into the detector within 45 s for further analysis.

Test parameters: injector temperature 200 °C, injection speed 125 μL/s. The initial temperature of the catch trap is 40 °C, the shunt rate is 10 mL/min, and the final temperature is 200 °C. The initial column temperature is 50 ℃, rising to 80 °C at 1 ℃/s and then rising to 250 °C at 3 °C/s, acquisition time 110 s, and FID detector temperature 260 °C.

#### 2.3.8. Data Statistics and Analysis

Each experiment was parallel three times. Excel was used to calculate the mean value and standard deviation, SPSS 17.0 was used for the statistical analysis of the data, and Origin 2021 was used for mapping.

## 3. Results and Discussion

### 3.1. Effect of Material-to-Oil Ratio and Reducing Sugar Addition on Sensory Evaluation of Zanthoxylum Seasoning Oil

The influence of material-to-oil ratio and reducing sugar addition on sensory evaluation of *Zanthoxylum* seasoning oil was shown in Figure 2. As shown in Figure 2A, with the increase in the material-to-oil ratio, the aroma and taste of *Zanthoxylum* seasoning oil increased at first and then decreased. The color was positively correlated with the material-to-oil ratio, indicating that the addition of enzymatic hydrolysate had a certain influence on the color of the oil during the thermal reaction. Taking all factors into consideration, three material-to-oil ratios of 1:3, 1:5 and 1:7 were selected for the orthogonal experiment. As shown in Figure 2B, with the increase in reducing sugar addition, the aroma of *Zanthoxylum* seasoning oil increased first and then decreased, but the taste and color of that were barely affected. In the initial phase of the reaction, with the increase in reducing sugar addition, the reaction process between reducing sugar and amino acid in the enzymatic hydrolysate was accelerated. When the addition amount was more than 2%, the aroma of *Zanthoxylum* seasoning oil gradually decreased, probably because the reaction between reducing sugar and amino acid in enzymatic hydrolysate had been saturated at this time, and the burnt aroma produced by the Maillard reaction might also have a slight effect on the aroma of *Zanthoxylum bungeanum*. Therefore, based on the overall scores, the reducing sugar additions of 1.5%, 2% and 2.5% were selected for orthogonal experiments.

### 3.2. Effect of Heating Temperature on Sensory Evaluation and Quality

The effect of heating temperature on sensory evaluation and quality was shown in Figure 3. As shown in Figure 3A, the peroxide value continued to increase, which was due to the accelerated oxidation of oils at high temperatures [21,22]. Moreover, the acid value decreased slightly, which was presumably due to the accelerated evaporation of water from the oil with the heating temperature increasing. As shown in Figure 3B, the fagaramide content decreased, which was because the amides in *Zanthoxylum* seasoning oil were destroyed at high temperatures, and its conjugated triene bond generated more stable derivatives by oxidation and hydration leading to a decrease in numb taste [23,24]. With the increase in heating temperature, the sensory of *Zanthoxylum* seasoning oil increased first and then decreased. When the temperature exceeded 120 °C, the color would be affected, which was due to the intensification of the Maillard reaction with the increase in temperature [25]. When the temperature exceeded 130 °C, the degree of numb taste decreased. By comprehensive consideration, the heating temperatures of 110, 120 and 130 °C were selected for orthogonal experiments.

### 3.3. Effect of Reaction Time on Sensory Evaluation and Quality

The effect of reaction time on sensory evaluation and quality was shown in Figure 4. As shown in Figure 4A, with the increase in the reaction time, the peroxide value increased, but the acid value was barely affected. As shown in Figure 4B, when the reaction time was longer than 25 min, the aroma of *Zanthoxylum bungeanum* weakened, which was presumably because the flavor substances generated by the Maillard reaction covered part of it. When the reaction time was longer than 20 min, the taste started to decline, which might be because the burnt flavor produced by the Maillard reaction had a certain covering effect on the taste of *Zanthoxylum seasoning oil* [26], or because the fagaramide was hydrolyzed and oxidized under continuous high temperature leading to the decrease in numb taste [27]. When the reaction time was too long (>25 min), the content of fagaramide started to decline, which was consistent with the taste change in the sensory evaluation. To sum up, the reaction time of 20, 25 and 30 min was selected for orthogonal experiments.

### 3.4. Orthogonal Experiments

According to the results of the single factor test, material-to-oil ratio (A), heating temperature (B), reaction time (C), and reducing sugar addition (D) were selected for the L9 (34) orthogonal experiment. The factor level table was shown in Table 2, and the orthogonal experiment design and results were shown in Table 3.

As shown in Table 3, the impact degree of four factors A, B, C and D on the sensory of *Zanthoxylum* seasoning oil was as follows: B (heating temperature) > D (reducing sugar addition) > A (material-to-oil ratio) > C (reaction time). After optimization, the optimal technology for *Zanthoxylum* seasoning oil was A_2_ B_1_ C_2_ D_2_ or A_2_ B_1_ C_3_ D_2_; the material-to-oil ratio was 1:5, the heating temperature was 110 °C, the reaction time was 25 or 30 min, and the reducing sugar addition was 2%. Under these conditions, the sensory score of *Zanthoxylum* seasoning oil based on the Maillard reaction prepared by the validation experiment was 8.7, with the peroxide value being 0.136 g/100 g, the acid value was 0.389 mg/g, and the *Zanthoxylum* amide was 2.70 mg/g.

### 3.5. Effect of Blending Ratio on Sensory Evaluation and Quality Indexes of Zanthoxylum Seasoning Oil

The effects of different blending ratios on the sensory evaluation, acid value, peroxide value and fagaramide of *Zanthoxylum* seasoning oil were shown in Table 4, and the correlation analysis among the indicators was shown in Table 5. As shown in Table 4, the blending ratio had a great effect on the overall score of *Zanthoxylum* seasoning oil. Cold pressing oil had a heavier bitter and numbing taste, while hot infused oil had a heavier aroma. With the increase in blending ratio, the color changed from black to dark green and then to yellow-green, and the bitter taste and numb taste in *Zanthoxylum* seasoning oil decreased gradually, and the sensory score increased significantly (*p* < 0.05). When the blending ratio was 1:7, the *Zanthoxylum* seasoning oil had the best sensory properties, at which time it had a strong pepper aroma, heavy numb taste and no bitter taste, and the color was yellow-green.

Peroxide value was an important index to identify the degree of oil deterioration [28] Compared to soybean oil, LZ and RJ had higher peroxide values, which might be due to the exposure to air during the soaking and pressing of the oil and *Zanthoxylum bungeanum* and the oil oxidation catalyzed by the high temperature during the hot dipping. The peroxide value of *Zanthoxylum* seasoning oil with each blending ratio had been tested to conform to national standards.

The acid value was determined as the free fatty acid produced by the hydrolysis of triglycerides, which was considered to be one of the main indicators of oil quality [29]. When the oil was oxidized and rancid, it would produce a “hala” taste, which could also cause cardiovascular diseases such as cirrhosis [30]. The acid value of LZ was higher, presumably due to the water being squeezed out of the *Zanthoxylum bungeanum* during pressing. The acid value of RJ was lower, which was due to the lower moisture content in the *Zanthoxylum bungeanum* cake after hydraulic pressure, while the residual moisture was evaporated at high oil temperatures. The acid value of each *Zanthoxylum* seasoning oil had been tested to conform to national standards.

Fagaramide was an important taste compound (numb taste) in *Zanthoxylum* seasoning oil and also provided a protective effect on the stomach [31]. The fagaramide content in LZ was significantly higher than that in RJ (*p* < 0.05), indicating that cold pressing could well retain the numb-taste substances in *Zanthoxylum bungeanum*. With the increase in blending ratio, the fagaramide content had an overall downward trend. According to the test, the content of fagaramide under different blending ratios exceeded the standard specified in DBS51/008-2019 (≥2 mg/g). It could be seen from Table 5 that there were strong correlations among the indicators. The sensory score was negatively correlated with acid value, peroxide value and fagaramide (*p* < 0.01), and the remaining indicators were positively correlated with each other (*p* < 0.01). Combining the blending ratios and the available data, the increase in acid value and peroxide value has a negative impact on the oil flavor, while the excessively numb taste would also affect consumer acceptance. Combining the acid value, peroxide value and sensory evaluation of *Zanthoxylum* seasoning oil with each ratio, the ratio of 1:7 for cold pressing oil and hot dipping oil (LH) was selected as the optimal ratio.

### 3.6. Sensory Evaluation and Quality Indexes Comparison of Zanthoxylum Seasoning Oil Based on Maillard Reaction, Fragrant Zanthoxylum Seasoning Oil and Zanthoxylum Seasoning Oil

The sensory evaluation of MH, LH and HL was shown in Figure 5. The acid value, peroxide value and fagaramide content were shown in Table 6. It could be seen from Figure 5 that the sensory evaluation of LH was the best, followed by MH and HL, which was mainly reflected in the aroma. LH had a more intense and persistent pepper aroma and heavier numb taste, while MH and HL have no significant differences in numb taste, which was consistent with the results of fagaramide determination. It could be seen from Table 6 that LH had a higher acid value, which might be caused by water entering the oil during the hydraulic, while MH and HL have no difference in acid value. Compared with HL, the peroxide values of MH and LH were higher. The higher peroxide value of MH might be due to the oil oxidation promoted by the high-temperature heating during the preparation of MH, while that of LH might be due to the exposure to air during the mixing and extraction of *Zanthoxylum bungeanum* and oil in the preparation of LZ.

### 3.7. Analysis of Fatty Acid Composition of Zanthoxylum Seasoning Oil Based on Maillard Reaction, Fragrant Zanthoxylum Seasoning Oil and Zanthoxylum Seasoning Oil

The fatty acid composition of MH, LH and HL was shown in Table 7. It could be seen from Table 7 that the content of unsaturated fatty acids in MH, LH and HL was relatively high, including 82.95% for MH, 83.76% for LH and 85.09% for HL. Fatty acids such as capric acid, ginkgolic acid and methyl linolenic acid were not detected in HL, but they were present in smaller amounts in MH and LH, which might be due to different processing techniques.

### 3.8. Analysis of Vitamin E and Sterol Content in Soybean Oil, Zanthoxylum Seasoning Oil and Fragrant Zanthoxylum Seasoning Oil

As could be seen from Table 8, the vitamin E content in DD was 0.1%, which was consistent with that (mean content 1052.6–1138.1 mg/kg) detected by Wen Yunqi et al. [32]. There was no significant difference between the vitamin E of LZ, RJ, HL and LH and that of soybean oil. The sterol contents of HL, LZ, and LH were 0.23%, 0.24%, and 0.23%, respectively, which were all higher than that of DD (0.16%). It was speculated that the sterol in *Zanthoxylum bungeanum* was partially dissolved in *Zanthoxylum* seasoning oil during cold pressing and hot dipping. The low sterol content of 0.14% in RJ might be due to the high temperature during the heating of soybean oil thus reducing the retention of sterols [33].

### 3.9. Analysis of Volatile Flavor Compounds of Zanthoxylum Seasoning Oil Based on Maillard Reaction, Fragrant Zanthoxylum Seasoning Oil and Zanthoxylum Seasoning Oil

The principal component analysis plots among MH, HL and LH samples were shown in Figure 6. It could be seen from Figure 6 that the variance contribution rate of the first principal component was 93.03%, that of the second principal component was 6.82%, and the total contribution rate of the two was 99.85%, which indicated that the graph could reflect the integrity of the odor data of the measured samples. The inter-sample identification index was 96, indicating that the samples were effectively distinguished from each other. There was no intersection among LH, MH and HL in Figure 6, and the spacing was large, indicating that there was a certain difference between the three groups of samples.

The Heracles II ultra-fast gas phase electronic nose was applied to test the volatile flavor compounds of LH, MH and HL. Combined with the search of the database, the volatile flavor components that might be contained in the samples were obtained (Table 9). It could be seen from Table 9 that 19 volatile flavor compounds were detected in LH, 16 in MH and 15 in HL, among which alcohols and alkenes were the main ones. The olefins content in volatile flavor compounds of the three kinds of *Zanthoxylum* seasoning oil was relatively high, 51.25% (LH), 42.55% (MH), and 55.83% (HL), respectively [34]. The relative contents of alcohols in the three kinds of *Zanthoxylum* seasoning oil were 29.73% (LH), 45.87% (MH), and 26.19% (HL), indicating that olefins and alcohols substances contributed more to the flavor of *Zanthoxylum* seasoning oil. As the flavor substance with the highest content among olefins, Limonene could effectively relieve cough and inhibit bacteria and was widely used in food as a spice. Linalool, as the flavor substance with the highest content among alcohols, was a characteristic component of *Zanthoxylum bungeanum* and had a strong influence on the numb taste [35]. It had floral, woody and fruity fragrances, which were often used in spices and had antibacterial, anti-inflammatory, hypnotic and other effects [36].

Except for olefins and alcohols, the contents of volatile flavor compounds in the three kinds of *Zanthoxylum* seasoning oils were in the following order: alkanes > aldehydes > ketones > lipids. With a special smell, P-cymene could be used as an intermediate in spices and pharmaceuticals, which also had cough relieving and expectorant effects, and was an aroma-active substance that distinguishes *Zanthoxylum* seasoning oils prepared from different origins [37]. Among the three kinds of *Zanthoxylum* seasoning oil, the relatively high contents of volatile flavor compounds were limonene, linalool, eucalyptol, n-pentane, α-Pinene, myrcene, phellandrene, which were basically consistent with the research results of [3,38,39]. Eucalyptol had a camphor and herbal flavor. α-Pinene had the smell of turpentine. Myrcene had the smell of balsam. α-Hyacinene had the smell of fresh citrus and pepper. α-Terpinolene had the smell of pine and citrus. These flavor compounds constituted the overall smell of *Zanthoxylum* seasoning oil, and the other flavor substances played a coordinating role in the overall aroma of that [40,41].

## 4. Conclusions

In this paper, the technology of Zanthoxylum seasoning oil based on the Maillard reaction was optimized by a single-factor test and orthogonal test. The Heracles II ultra-fast gas phase electronic nose was applied to test the volatile flavor compounds of Zanthoxylum seasoning oil based on the Maillard reaction, fragrant Zanthoxylum seasoning oil and Zanthoxylum seasoning oil, as well as to perform quality analysis. The results showed that the optimal technology of MH was as follows: the material-to-oil ratio was 1:5, the heating temperature was 110 °C, the reaction time was 25 or 30 min, and the addition of reducing sugar was 2%. Under these conditions, the sensory score of MH was 8.7, the peroxide value was 0.136 g/100 g, the acid value was 0.389 mg/g, and the fagaramide content was 2.70 mg/g. The obtained LH had a strong aroma, heavy numb taste, no bitter taste, and long retention time. Sixteen volatile flavor compounds were detected in MH, 19 in LH and 15 in HL. Among the three kinds of Zanthoxylum seasoning oil, the contents of limonene, linalool, Eucalyptol, n-pentane α-Pinene, myrcene, and phellandrene were more abundant, which indicated that olefins and alcohols contributed more to the overall flavor. This study could provide some theoretical basis for the product development of Zanthoxylum seasoning oil manufacturers.

## Figures and Tables

**Figure 1 foods-12-02173-f001:**
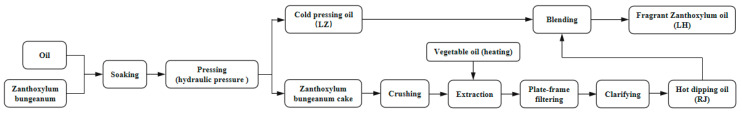
Process flow chart of fragrant *Zanthoxylum seasoning oil*.

**Figure 2 foods-12-02173-f002:**
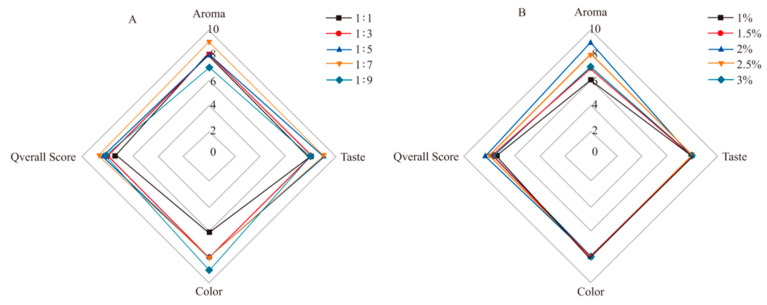
(**A**)The sensory evaluation of Zanthoxylum seasoning oil under different material oil ratio and (**B**) the sensory evaluation of Zanthoxylum seasoning oil with different amount of reducing sugar.

**Figure 3 foods-12-02173-f003:**
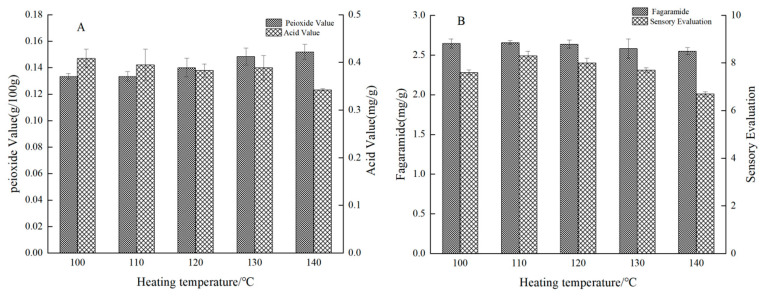
(**A**,**B**) Effect of different heating temperature on peroxide value, acid value, fagaramide of *Zanthoxylum* and sensory evaluation.

**Figure 4 foods-12-02173-f004:**
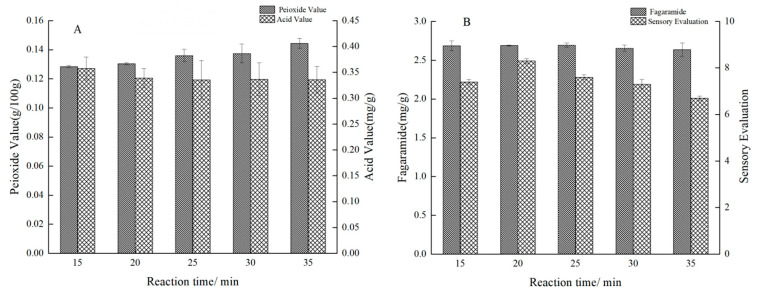
(**A**,**B**) Effect of different reaction time on peroxide value, acid value, fagaramide of *Zanthoxylum* and sensory evaluation.

**Figure 5 foods-12-02173-f005:**
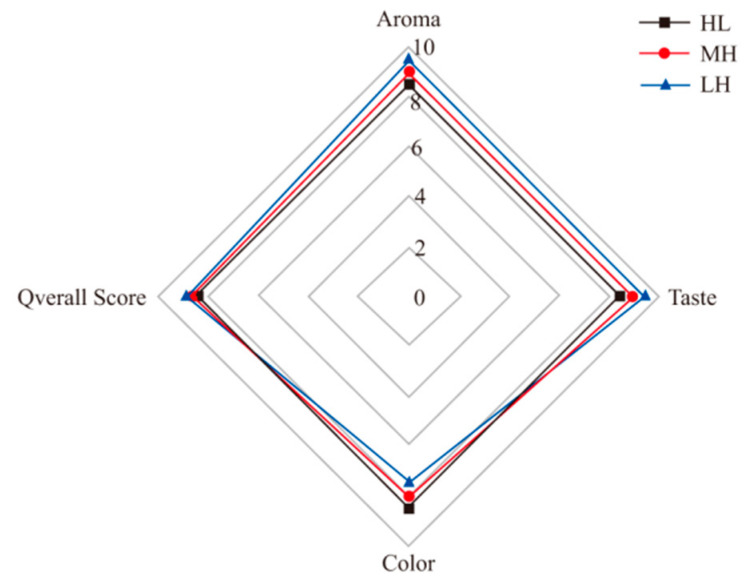
Sensory evaluation of the *Zanthoxylum* seasoning oil based on Maillard reaction (MH), fragrant *Zanthoxylum* seasoning oil (LH) and *Zanthoxylum* seasoning oil (HL) sample.

**Figure 6 foods-12-02173-f006:**
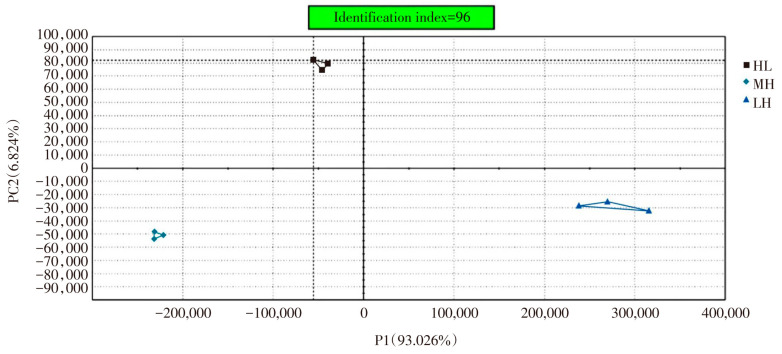
**PCA** analysis diagram of the *Zanthoxylum* seasoning oil based on Maillard reaction (MH), fragrant *Zanthoxylum* seasoning oil (LH) and *Zanthoxylum* seasoning oil (HL) sample.

**Table 1 foods-12-02173-t001:** Sensory evaluation standard of *Zanthoxylum seasoning oil*.

Score	Aroma (0~10 Points)	Taste (0~10 Points)	Color (0~10 Points)
0–3	No obvious pepper aroma	No significant numb taste	Light yellow, cloudy
4–6	Characteristic pepper aroma, no odor	Obvious numb taste, retention time short; heavy numb taste and bitter taste	Dark color, a little impurity
7–10	Strong pepper aroma, no odor	Heavy numb taste, no bitter taste, retention time long	Brownish yellow, good transparency

Note: Overall score = aroma score × 0.3 + taste score × 0.4 + color score × 0.3.

**Table 2 foods-12-02173-t002:** Factor level table of orthogonal experiment.

Level	A. Material-to-Oil Ratio	B. Heating Temperature /°C	C. Reaction Time/min	D. Reducing Sugar Addition/%
1	1:3	110	20	1.5
2	1:5	120	25	2
3	1:7	130	30	2.5

**Table 3 foods-12-02173-t003:** Orthogonal experimental design and results.

Experiment Number	A	B	C	D	Sensory Score
1	1	1	1	1	7.4
2	1	2	2	2	7
3	1	3	3	3	7.1
4	2	1	3	2	8.7
5	2	2	1	3	7
6	2	3	2	1	7.7
7	3	1	2	3	7.7
8	3	2	3	1	6.7
9	3	3	1	2	8.3
k1	7.17	7.93	7.43	7.27	
k2	7.8	6.9	7.47	8	
k3	7.57	7.7	7.47	7.27	
*R*	0.63	1.03	0.04	0.73	

**Table 4 foods-12-02173-t004:** Sensory evaluation and physicochemical indexes of Zanthoxylum seasoning oil with different proportions.

Blending Ratio	Sensory Score	Acid Value	Peroxide Value	Fagaramide
LZ(1:0)		2.27 ± 0.13 a	0.19 ± 0.002 a	30.70 ± 0.3 a
RJ(0:1)		0.51 ± 0.02 g	0.11 ± 0.007 i	1.78 ± 0.02 j
1:1	5.4 h	1.33 ± 0.03 b	0.16 ± 0.005 b	13.63 ± 0.15 b
1:2	5.4 h	0.89 ± 0.07 c	0.16 ± 0.001 b	9.29 ± 0.10 c
1:3	5.8 g	0.79 ± 0.009 d	0.15 ± 0.001 c	7.96 ± 0.13 d
1:4	6.9 f	0.76 ± 0.006 d	0.15 ± 0.002 d	6.32 ± 0.15 e
1:5	7.6 e	0.65 ± 0.02 def	0.14 ± 0.002 de	4.69 ± 0.17 f
1:6	8.6 c	0.69 ± 0.08 e	0.14 ± 0.002 ef	4.26 ± 0.06 g
1:7	8.9 a	0.58 ± 0.002 efg	0.14 ± 0.001 f	4.29 ± 0.12 g
1:8	8.8 b	0.59 ± 0.03 efg	0.13 ± 0.002 g	3.67 ± 0.04 h
1:9	8.2 d	0.56 ± 0.007 fg	0.12 ± 0.002 h	3.41 ± 0.11 i
1:10	8.4 d	0.56 ± 0.009 fg	0.11 ± 0.004 i	3.30 ± 0.03 i

Note: Different lowercase letters in the same column indicate a significant difference (*p* < 0.05).

**Table 5 foods-12-02173-t005:** Correlation analysis among indicators.

	Sensory Score	Acid Value	Peroxide Value	Fagaramide
Sensory Score	1			
Acid Value	−0.81 **	1		
Peroxide value	−0.845 **	0.844 **	1	
Fagaramide	−0.894 **	0.995 **	0.856 **	1

Note: ** is *p* < 0.01, indicating a significant difference.

**Table 6 foods-12-02173-t006:** Sensory evaluation and quality indexes comparison of MH, LH and HL.

Samples	Acid Value/(mg/g)	Peroxide Value/(g/100 g)	Fagaramide/(mg/g)
MH	0.389 ± 0.002	0.14 ± 0.002	2.70 ± 0.14
HL	0.342 ± 0.003	0.09 ± 0.001	2.62 ± 0.15
LH	0.58 ± 0.002	0.14 ± 0.002	4.29 ± 0.16

**Table 7 foods-12-02173-t007:** Fatty acid composition of MH, LH and HL.

Fatty Acid Composition/%	MH	LH	HL
Capric acid C10:0	0.51 ± 0.02	0.72 ± 0.04	-
Myristic acid C14:0	0.08 ± 0.01	0.09 ± 0.01	0.08 ± 0.01
Palmitic acid C16:0	11.51 ± 0.05	11.08 ± 0.03	11.18 ± 0.21
Palmitoleic acid C16:1	0.12 ± 0.01	0.16 ± 0.02	0.10 ± 0.01
Heptadecanoic acid C17:0	0.10 ± 0.01	0.10 ± 0.01	0.08 ± 0.01
Ginkgolic acid C17:1	0.04	0.07 ± 0.01	-
Stearic acid C18:0	4.11 ± 0.03	4.01 ± 0.01	3.30 ± 0.21
Oleic acid C18:1	20.90 ± 0.1	20.53 ± 0.04	21.15 ± 0.41
Linoleic acid C18:2	52.87 ± 0.12	53.14 ± 0.05	55.16 ± 1.34
Methyl linolenic acid C18:3 n6	0.42 ± 0.03	0.67 ± 0.01	-
α-Linolenic acid C18:3 n3	8.43 ± 0.07	9.03 ± 0.03	8.68 ± 0.53
Arachidic acid C20:0	0.74 ± 0.03	0.24 ± 0.01	0.15 ± 0.01
Arachidonic acid C20:1	0.17 ± 0.01	0.16 ± 0.01	0.12 ± 0.01

**Table 8 foods-12-02173-t008:** Vitamin E and sterol content in soybean oil, Zanthoxylum seasoning oil, cold pressed oil, hot dipped oil and fragrant *Zanthoxylum seasoning oil*.

	Vitamin E/%	Sterol Content/%
Soybean Oil	0.1	0.16
HL	0.11	0.23
LH	0.1	0.23

**Table 9 foods-12-02173-t009:** Qualitative analysis of aroma components of fragrant *Zanthoxylum* seasoning oil, *Zanthoxylum* seasoning oil from Maillard and *Zanthoxylum seasoning oil*.

Order Number	Possible Flavor Substances	Flavor Information	Volatile Flavor Compounds/%		
			LH	MH	HL
1	n-pentane	Faint mint scent	12.35	4.26	11.48
2	Cyclopentane		0.49	-	-
3	N-Hexane		0.41	-	-
4	2-Methylhexane		0.28	-	-
5	Vinyl acetate	Sweet ether fragrance	065	-	-
6	Ethyl acetate	Fruity aroma	0.26	-	-
7	delta-Nonalactone	Cocoa, cream, nut flavor	-	0.99	
8	α-Pinene	Turpentine flavor	2.72	3.8	7.53
9	trans-2-Heptenal		1.71	1.43	2.37
10	Myrcene	Balsam smell	20.07	10.83	12.41
11	α-Hyacinene	Fresh citrus and pepper	6.53	58.74	8.51
12	P-cymene	The smell of damp and moldy dishcloth	0.44	0.69	0.88
13	Limonene	lemon	19.7	18.84	23.04
14	α-Isoterpinene	Pine smell, citrus smell	2.23	2.86	4.34
15	eudesmol	Camphor flavor	1.18	2.05	2.8
16	Linalool	Fruity, woody, floral	28.55	43.82	22.75
17	Ethanol	Irritating odor	-	-	0.64
18	2,3-Pentanedione	Cream, caramel, nutty smell	0.36	-	-
19	3,5-Octadiene-2-one	Fresh grass taste	0.78	0.85	
20	Acetone		-	-	1.1
21	(2 E, 4 E)-2,4-octadienal		0.57	0.69	0.53
22	Citronellal	Fresh green citrus, wood flavor	0.71	0.95	0.99
23	Hexanal	Fatty taste, green grass taste, apple taste	-	0.92	0.64

Note: “-” means the volatile flavor substance was not included.

## Data Availability

The data presented in this study are available on request from the corresponding author.

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
