# Peer review of "Study on Production Technology and Volatile Flavor Analysis of Fragrance Zanthoxylum Seasoning Oil"

_foods, 2023, doi:10.3390/foods12112173_

Round 1
Reviewer 1 Report
Manuscript # foods-2333313 Study on production technology and volatile flavor analysis of fragrance Zanthoxylum seasoning oil
I am having hard time finding the rationale for this study, and the experimental design. I would strongly recommend authors to clearly state the justification of this study. I am not following the fact that cold press oil being heated and I am not sure if conditions for Maillard reaction should be considered in cold press oil, as MR is typically reacted under heated condition. Also, sensory evaluation method should be more clearly stated in the matierals and method section, such as sample serving protocols, palate cleansing method, sample evaluation method, resting time etc. Volatile compound analysis -- please provide more specific condition for analysis method so that readers can reproduce this study.
Author Response
Dear Editor,We are pleased to answer the questions of the reviewers’ and the manuscript has also been extensively revised according to the comments.
This study is to study the effect of Maillard reaction and cold pressed on the flavoring of Zanthoxylum flavoring oil. One is the Maillard reaction under heat, the other is the cold pressed, and there is no cross between the two.
Sensory evaluation method has been redescribed:Appropriate amount of samples were taken in the beaker and the color was observed under natural light.The product color was evaluated according to Table 1.Heat a beaker in a water bath to 50℃, stir it quickly with a glass rod,the product odor was evaluated according to Table 1.Rinse your mouth with warm water, and taste,the taste of the product was evaluated according to Table 1 and the tasted interval of each product was 5min
The analysis method for volatile flavor compounds has been modified as follows:The volatile flavor compounds of Oil samples from Maillard reaction(MH),fragrant Zanthoxylum seasoning oil (LH) and Zanthoxylum seasoning oil (HL) of the volatile flavor compounds were taken by Heracles II ultra-fast gas phase electronic nose.Briefly,2g of the oil sample was placed in 20mL headspace bottle and water bath for 24min(55℃).suck up 5000μL with an injection needle gas, injected into the detector within 45 seconds for further analysis.Each sample was manually injected, for which three parallel experiments was performed.

Reviewer 2 Report
Comments
In the present work, Maillard reaction and cold pressed process were developed and optimized for improving the aroma and taste of Zanthoxylum bungeanum seasoning oil. The evaluations in sensory, peroxide/acid/fagaramide values, fatty acid/aroma compositions, and vit. E/sterol contents were exploded for optimizing parameters of developed processes. The optimized MH base on the Maillard reaction reveled more intense and persistent aroma. The taste of fragrant Z. bungeanum seasoning oil (LH) was the best.
The present results provide design informations for manufactures. The topic is relevant in the field of “Foods”.
1. There are some typing and grammar errors that should be revised carefully (including the References, Tables and Figures). Please revise and mark the correction carefully.
2. Please consist the units.
3. Please provide the moisture of matrix.
4. The further weakness of present manuscript is the incomplete methods and inconsistent contents that bring the confuse. Improving the procedure clearness and result comprehension are highly suggested.
5. Please provide the yields in each step for proving the clear procedure comprehension and highlighting the goal of product development.
Ex: vegetable oil ↔ soybean oil
dried green pepper ↔ ground prickly ash
↔ enzymatic hydrolyzate of degreasing Zanthoxylum bungeanum
......mixed with Zanthoxylum seasoning oil in proportion
glucose ↔ sugar
mg/g ↔ g/100g (in Figs 3 & 4) (omit the units in Table 6)
omit the process obtained (in some Figs and Tables)
6. By using the reported Maillard reaction and cold pressing procedure to improve the qualities of Z. bungeanum seasoning oil are the subject of present work. The other reported methods, such as acid-catalyzed esterification, ultrasound-assisted, microwave-assisted, and supercritical CO2 assisted methods, were recommended to be a control for the comparison.
Author Response
Dear Editor,We are pleased to answer the questions of the reviewers’ and the manuscript has also been extensively revised according to the comments.
- There are some typing and grammarerrors that should be revised carefully (including the References,Tables and Figures).Please revise and mark the correction carefully.
These questions have been revised,such as:
Spaces are added between numeric values and units of measurement;
An unreferenced chart was added to the study;
Some units in the Figs and Tables has been supplemented, and et al.
Some units in the Table 6 has been supplemented.
- Pleaseconsistthe units.
Some units in the Table 6 has been supplemented.
- Please provide themoistureof matrix.
The matrix moisture content was9.6%
- Thefurther weakness of present manuscript is theincomplete methods and inconsistent contents that bring the confuse. Improving the procedure clearness and result comprehension are highly suggested.
Some method descriptions have been modified and marked in the paper
5.Please provide the yields in each stepforproving theclearprocedure comprehension and highlighting the goal of productdevelopment.
Ex:vegetable oil ↔soybean oil
dried green pepper↔ ground prickly ash↔ enzymatic hydrolyzate of degreasing Zanthoxylum bungeanum
......mixed withZanthoxylum seasoning oil in proportion
glucose↔sugar mg/g ↔ g/100g(in Figs 3 & 4)(omit theunitsin Table 6)
omit the process obtained(in some Figs and Tables)
The total volume of each mixed peppercorns seasoning oil has been replenished“Total volume of oil per serving was 100ml”
Sample dosage for other experimental steps has been indicated,such as 2.3.3“(the total weight of enzymatic hydrolysate and Zanthoxylum seasoning oil was 48 g)”,2.3.2“ 4500 g dried green pepper were weighed and crushed through a 10 mesh sieve”and et al.
Some units in the Figs and Tables has been supplemented
- By using the reported Maillard reaction andcold pressing procedure to improvethe qualities of bungeanumseasoning oil are the subject of present work. The otherreported methods, such asacid-catalyzed esterification, ultrasound-assisted, microwave-assisted, and supercritical CO2 assisted methods, were recommended to be a control for the comparison.
Thank you for your suggestion. We have supplemented it in the preface section.
Most of Zanthoxylum seasoning oil sold in the market today was made by directly soaking peppers in vegetable oil. However, in the previous research on the processing of Zanthoxylum seasoning oil, it was found that the numb taste and aroma components would be partially lost during its processing and storage[8], both of which were important indicators for quality evaluation of Zanthoxylum seasoning oil. Therefore, how to increase the aroma of Zanthoxylum seasoning oil and its persistence of numb taste was particularly important. At present, apart from the oil immersion method, the main methods for extracting Zanthoxylum oil from Zanthoxylum include microwave assisted extraction [9], ultrasonic assisted extraction [10], and supercritical CO2 extraction. But the operation of microwave assisted extraction is relatively complex, and the ultrasonic assisted extraction method requires the addition of organic solvents, which can easily cause solvent residues. The supercritical CO2 extraction method has high safety, no solvent residues, and less loss of flavor substances in Zanthoxylum, but the cost is high.In order to increase the aroma of Zanthoxylum seasoning oil, the experiment mainly adopts two ways, one is to add Maillard reaction, the other is to use cold pressing technology.

Author Response
Dear Editor, we are pleased to answer the questions of the reviewers’ and the manuscript has also been extensively revised according to the comments.
1.There is no reference to figure 1 in the text.
This reference was added in 2.3.2.“Process flow chart of fragrant Zanthoxylum seasoning oil was shown in Figure 1”.
2.There should be a space between the numeric value and the unit of measure – page: 82, 83, 88, 107, 115, 118, 121, 131, 133, 134, 136, 137, 138, 141, 142, 143, 145, 146, 147, 193, 194, 195, 197, 198, 199, 200, 201, 274, 275
Thanks for your reminding,this question has been revised in the study.
